# Grassland Resilience to Woody Encroachment in North America and the Effectiveness of Using Fire in National Parks

Han Ling, Guangyu Wang *, Wanli Wu, Anil Shrestha and John L. Innes

Faculty of Forestry, University of British Columbia, 2424 Main Mall, Vancouver, BC V6T 1Z4, Canada;
linghan0@mail.ubc.ca (H.L.); wanli.wu@ubc.ca (W.W.); anil.shrestha@ubc.ca (A.S.); john.innes@ubc.ca (J.L.I.)
* Correspondence: guangyu.wang@ubc.ca

**Abstract:** The grasslands of North America are threatened by woody encroachment. Restoring historical fire regimes has been used to manage brush encroachment. However, fire management may be insufficient due to the nonlinear and hysteretic responses of vegetation recovery following encroachment and the social–political constraints affecting fire management. We synthesized the fire thresholds required to control woody encroachment by typical encroaching species in North America, especially the Great Plains region, and identified the social–political constraints facing fire management in selected grassland national parks. Our synthesis revealed the resistance, hysteresis, and irreversibility of encroached grasslands using fire and emphasized the need for a combination of brush management methods if the impacts of climate change are to be addressed. Frequent fires alone may maintain grassland states, reflecting resistance. However, high-intensity fires exceeding fire-mortality thresholds are required to exclude non-resprouting shrubs and trees, indicating hysteresis. Fire alone may be insufficient to reverse encroachment by resprouting species, exhibiting reversibility. In practice, appropriate fire management may restore resistant grassland states. However, social–political constraints have restricted the use of frequent and high-intensity fires, thereby reducing the effectiveness of management actions to control woody encroachment of grasslands in national parks. This research proposes a resilience-based framework to manage woody encroachment in grassland national parks and similar protected areas.

**Keywords:** woody encroachment; fire thresholds; resilience; hysteresis; national park





## 1. Introduction

Woody encroachment, characterized by an increase in the abundance and dominance of trees and shrubs in grasslands, poses a significant threat to the survival of grassland ecosystems in North America [1]. The loss of these ecosystems will lead to the loss of essential ecosystem services [2]. The Great Plains region has experienced the highest woody encroachment rate among all ecoregions in North America, with a 1–2% annual increase in woody species cover due to the combined effects of altered fire regimes, drought, and grazing [2,3]. Before European settlement, frequent, low-intensity fires prevented the establishment of woody species and kept fire-adapted grasslands open, which in turn maintained the natural fire regimes. Fire exclusion interrupted this self-reinforcing feedback and resulted in woody encroachment. To address this issue, the Integrated Brush Management System (IBMS) was developed as a management framework that uses fire as well as mechanical, chemical, and biological treatments to reverse woody encroachment [4]. Among the methods used in the IBMS, fire is the most widely adopted due to its effectiveness in causing the mortality of shrub and tree canopies across extensive spatial areas while keeping long-term treatment expenses low.

Fire management that replicates historical fire regimes has been used to control woody encroachment [5]. However, fire management may not always be sufficient to reverse woody encroachment, and there are examples of continued management challenges [6].

Social–political constraints also limit the effectiveness of fire treatments for woody encroachment [7,8]. A better understanding of the recovery trajectories of grasslands after woody encroachment, the fire thresholds and efforts for fire management, and the social–political constraints on prescribed fires may help fire managers set clear and reachable management objectives in encroached grasslands.

We aimed to answer two research questions: (1) Could wildland fire reduce the woody encroachment of grasslands in North America, especially in the Great Plains region? We examined both prescribed fires and wildfires. The National Park Service defines wildland fires as including both unplanned wildfires caused by lightning or other natural causes and planned prescribed fires intentionally ignited by park managers to meet management objectives. We summarized encroaching species, fire management methods, and their impacts on woody encroachment through a review of published work. Ecological resilience is the amount of disturbance a system can absorb before shifting to an alternative state with new structures, functions, and processes [9–11]. We used this concept to detect the amount of fire intervention that can shift encroached grasslands back to the desired grassland state. (2) What are the social–political constraints limiting the effectiveness of brush management using fire in grassland national parks in Canada and the USA? We selected nine grassland national parks in Canada and the USA with a range of vegetation states to identify the most challenging social–political constraints facing the application of fire management. Our findings may help predict when fire management has potential or when it might fail in brush management.

## 2. Materials and Methods

We reviewed published work to identify the critical fire thresholds for major woody encroaching species in North America, using nine grassland national parks as case studies to summarize the social–political constraints limiting the effects of fire on reversing woody encroachment. For the review, we used three terms for the search strategy, including woody encroachment, fire, and North America. We used keywords in the topic ("woody encroachment" or "shrub encroachment", or "woodland encroachment" or "shrubland transition" or "woodland transition") and ("Great Plains" or Canada or US or USA or Mexico or "North America") and fire*. The first search generated 147 results in the Web of Science and 106 results in Scopus after excluding the literature from outside North America. A total of 253 references were imported into Covidence, and 159 studies were screened after the removal of 94 duplicates. We finally included 67 papers after screening titles and abstracts and reviewing full texts. The ecosystems in these studies were all experiencing woody encroachment, and fire was being used as a brush management method. The data extracted from these 67 papers covered encroaching species of shrubs and trees, fire characteristics and thresholds used for managing woody encroachment, and fire impacts on woody encroachment with their biological and physical attributes. We used resilience as a tool to qualitatively assess the effectiveness of fire on woody encroachment under the three resilience levels, including resistance, hysteresis, and irreversibility. The search results were limited to using resilience or related concepts as the fourth keyword. Publications defining resilience concepts were collected separately.

For the case studies, we selected five Canadian and four U.S. national parks. We included all five national parks in Canada that encompass grassland ecosystems: Grasslands National Park, Elk Island National Park, Riding Mountain National Park, Prince Albert National Park, and Waterton Lakes National Park. We selected four parks in the USA National Park System (within the Northern Great Plains Network and Southern Great Plains Network) within the Great Plains region: Badlands National Park, Lyndon B. Johnson National Historical Park, Chickasaw National Recreation Area, and Lake Meredith National Recreation Area. These parks were selected based on the presence of grassland ecosystems, the presence of representative encroaching species, the severity of woody encroachment, and the availability of research materials and data.

We identified 18 government assessment reports and management plans for these parks from the databases of Parks Canada (https://www.pc.gc.ca/en/index, accessed on 16 January 2023) and the US National Park Service (https://www.nps.gov/index.htm, accessed on 16 January 2023). We classified these parks into three vegetation states based on the encroaching conditions and management goals proposed by park managers in their management plans: (1) State zero, intact grassland: a well-preserved grassland state that managers have been monitoring but have not made any plans for reversing the encroachment. (2) Alternative state one, transition to shrublands: the encroaching species identified by managers are mainly shrubs, and managing the shrub encroachment is a stated objective in the management plan. (3) Alternative state two, woodland transitions: managers plan to control encroaching woody species, such as ponderosa pine (*Pinus ponderosa*), Ashe juniper (*Juniperus ashei*), Virginian juniper (*Juniperus virginiana*), honey mesquite (*Prosopis glandulosa*), and aspen (*Populus tremuloides*). In some parks, certain woody species are classified as shrub encroachers if they are less than 3m, and we employed identical definitions to those used in the park's management plan. Finally, we summarized the fire regimes, specific fire treatments, and the challenges of reversing woody encroachment in these parks.

## 3. Results

### 3.1. Ecological Resilience of Grasslands to Woody Encroachment

3.1.1. Woody Encroachment in the Great Plains of North America

The grasslands in the Great Plains region have been experiencing the most rapid woody encroachment of all regions in North America, losing 1–2% of grassland cover per year [2,3,12]. There are diverse grassland ecosystems of shortgrass and tallgrass prairies in the Great Plains region. The region is distinguished by relatively little topographic relief, grasslands, a paucity of forests, and by having a subhumid to semiarid climate [13]. The temperature decreases from south to north and the rainfall increases from west to east, driving changes in the composition of the vegetation [14,15]. Shortgrass prairies with grass shorter than 20 cm, occur in the drier western region and in the rain shadow of the Rocky Mountains. Tallgrass prairies, with grass higher than 200 cm, occupy the wetter eastern region. The biophysical settings affect the disturbance regimes of fire and grazing, and human activities influence the development and maintenance of the grassland ecosystems of the Great Plains region [16,17].

Species contributing to woody encroachment vary across the ecoregions within the Great Plains [12]. For example, in the Canadian prairies, shrubs, aspen, and lodgepole pine (*Pinus contorta*) have encroached into the native grasslands of foothills and parklands and the tallgrass prairie ecoregion due to the elimination of both fire and bison [18–20]. In the Northern Great Plains, Virginian juniper and Rocky Mountain juniper (*Juniperus scopulorum*) have been the most widespread encroachers with moderate and substantial encroachment rates [1]. In the Central Great Plains, oak (*Quercus* spp.) has encroached into the tallgrass prairie. Eastern red cedar and various shrub species encroached into the mixed-grass prairie. Juniper (*Juniperus* spp.) has proliferated in the tall- and mixed-grass prairies [21]. In the Southern Great Plains, honey mesquite (*Prosopis glandulosa*) and juniper have encroached into the semiarid grasslands due to fire suppression [22,23].

The encroaching shrubs and trees that were cited more than twice in the published studies are summarized in Figure 1. For shrubs, mesquite (*Prosopis* spp.), rough-leaved dogwood (*Cornus drummondii*), and smooth sumac (*Rhus glabra*) were the most studied encroaching shrubs [24–32]. Juniper (*Juniperus* spp.), pinyon (*Pinus* spp.), and oak (*Quercus* spp.) were the most studied encroaching trees [33–35]. The tall mesquite was also defined as encroaching trees in some studies. *Cornus drummondii, Rhus glabra*, and *Prunus americana* are clonal and resprouting shrubs [29]. Based on the encroaching species and the percent of woody cover, the encroached states of grasslands can be mainly defined as shrubland and woodland transitions.

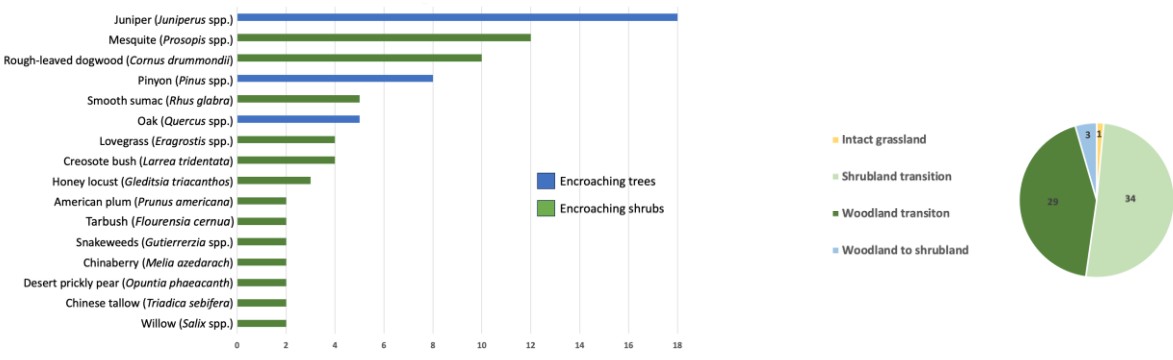

**Figure 1.** Encroaching species, encroached states of grasslands, and the number of times they were cited in the published studies.

### 3.1.2. Brush Management in the Great Plains Region

Brush management is a set of specific management methods aimed at reversing woody encroachment and restoring the grassland states. It involves the removal, reduction, or manipulation of target woody plants while protecting desired species [36,37]. The goal is to achieve a desired plant community based on the species composition, structure, density, and canopy cover or height. To achieve long-term brush management objectives, managers need to consider the advantages and disadvantages of different treatment methods and implement a combination of methods. One such approach is the Integrated Brush Management System (IBMS), which incorporates various treatment methods from a multi-use and long-term perspective [4]. The IBMS cycle involves setting management objectives, outlining potential treatments, applying treatments, monitoring and evaluating, and implementing additional follow-up treatments if necessary. However, the effectiveness of treatments can vary across landscapes, as shown by the limited long-term effects of current brush management methods in the Southern Great Plains region [38].

Brush management treatments include mechanical, chemical, biological, and cultural methods. Mechanical methods involve removing the top growth or the entire plant using equipment such as hand tools, haying, or heavy machinery. Removing both the root system and the top can result in long-term control of resprouting species but involves significant soil disturbance. Chemical methods, such as herbicides, are often used for broadcast applications or individual plant treatment. However, these methods have limitations in that they cannot remove the seeds that enable re-encroachment and their application over large areas can be costly [39]. Biological treatments, such as goat browsing, can reduce the seedlings and the early stages of encroachment, but controlling these treatments can be challenging. Cultural methods include prescribed fires and wildfires, targeted grazing, and range seeding.

Fire is an effective and commonly used method that can manage the encroachment process from seed dispersal to the mature woodland stage. Compatible with wildlife habitat requirements, fire has fewer impacts on the soil, lower treatment costs, and is feasible at larger spatial scales than other methods [40]. The characteristics of fire regimes have been used in managing woody encroachment. Fire regimes are the spatial and temporal characteristics of the fire event patterns affecting a particular landscape in space and through time [41]. Spatial attributes include fire location and position, fire size and shape, refugia (places that survived while the surrounding area was burned), position in the landscape relative to the topography and other disturbances, and mortality patterns in the disturbed area. Temporal attributes include the fire frequency (number of fire events per period), mean return interval (average number of years between successive fire events), fire recency (years after a burning), probability (the probability of a fire in any given year), predictability (whether the fire event is regular or sporadic), and rotation period (mean time required to disturb an area equivalent to the study area). Fire magnitude refers to the

direct and indirect impact of fire events, including the physical force (fire intensity) and indirect influences on vegetation and the environment (fire severity).

The fire type, fire interval, fire recency, seasonality, fire intensity, and fire severity were used for managing woody encroachment in the selected studies (Figure 2). The fire types include prescribed fire, wildfire, and wildland fire (prescribed fire and wildfire). Each study focused on one or multiple characteristics of the fire regime. The fire return interval/fire frequency was most studied in these fire regimes. Fire frequency, intensity, and severity were used to detect the fire thresholds and restore the encroached grasslands. Fire seasonality was studied to achieve high fire intensity. Fire recency was often used to detect long-term fire impacts on woody encroachment. Single brush management methods may not be sufficient to reverse woody encroachment; therefore, other brush management methods, such as herbicide, mechanical thinning, or cattle grazing, were also widely used and studied.

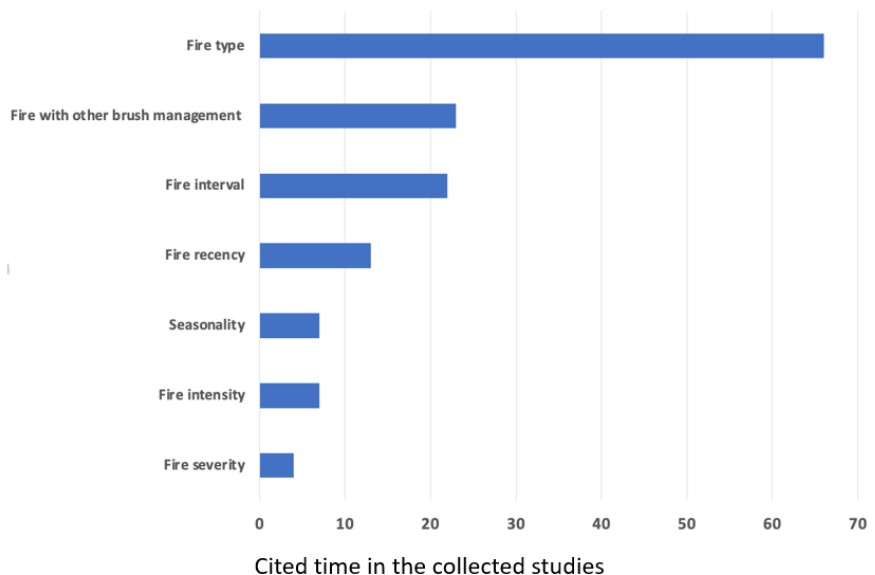

**Figure 2.** The fire regimes used for controlling woody encroachment in the selected studies.

Woody encroachment can be affected by many environmental (climatic, topo-edaphic, and disturbance) influences, human variables (urbanization, grazing, land management, and fire regimes changed by human activities), and brush management (mechanical, chemical, biological, and fire methods). The effects of these drivers may change at different spatial scales and landscape contexts [42]. For example, drought at a regional scale, as well as grazing and fire at the landscape and local scales, are the main ecological drivers in the Great Plains region [2,43].

In eight collected studies, climate variables, including severe water limitation and drought, precipitation frequency intensity, and temperature, were included to study the impacts of climate-fire interaction on the vegetation cover, biomass, ecohydrology, and fire severity The impacts were site-specific. For example, increased precipitation caused increased deep-water infiltration and availability, favoring woody species [44]; however, grass species also decreased in deep water availability by high transpiration, decreasing woody encroachment [45]. In the published studies, drought and severe water limitation conditions caused plant mortality of all species but favored drought-tolerance shrub encroachment [46]; while the precipitation frequency and intensity reduced the woody encroachment [29].

### 3.1.3. The Resilience of Grasslands: Resistance, Hysteresis, and Irreversibility

We used the resilience levels to evaluate the amount of fire or other management efforts a grassland needs to shift from an encroached state back to the previously intact grassland state. Changes in vegetation, wildlife, soil, habitat, and hydrological functions

were used to quantify the fire impacts on reversing the shrubland and woodland transitions in the published studies (Table 1). Vegetation indicators, especially vegetation cover and community composition, were most widely evaluated in these studies.

**Table 1.** Resilience indicators: fire impacts on woody encroachment in selected studies.

| Fire Impacts (Cited Time) | | Indicators |
| --- | --- | --- |
| Biological attributes | Vegetation (50) | Vegetation and land use cover, biomass, community composition, abundance, diversity, comparative ecohydrology of trees, shrubs, grass, survival probability, seed removal, regeneration, reestablishment, regrowth, flowering status, plant type competition, spatial patterns. |
| | Wildlife (5) | Richness, occurrence, survival rates, habitat selection, and nest survival. |
| Physical attributes | Soil (8) | Carbon and nitrogen and relative contribution from grasses and shrubs, strength of soil water repellency, soil moisture, particle-size distribution, decomposition rates of leaf litter, aeolian activity. |
| | Habitat (3) | Habitat quality, forage species, landscape patterns of habitat. |
| | Hydrological functions (7) | Soil hydrologic properties: soil water repellency, hydraulic conductivity, relative infiltration, runoff and rainfall, sediment. |

Resistance, hysteresis, and irreversibility were defined as the three levels of resilience [9,27,47]. A given ecosystem has resistance with a high level of resilience when it can recover to the previous state after removing the sudden discreet disturbance or restoring the pre-transition conditions under a continuous disturbance. The "path back" is the same as the "path forward", with a linear trajectory and unchanged thresholds [11]. For example, woody species have been encroaching into the grassland due to fire suppression, but the grass recovered after restoring the historical fire regime via prescribed fires [48,49]. These grasslands showed resistance to the initial woody encroachment. We assessed the shrubland and woodland transitions as resistance if the resilience indicators (Table 1) recovered under the historical fire regimes.

For a disturbed ecosystem with hysteretic responses, eliminating the external forcing or returning driving variables to their pre-transition levels is not efficient in reversing the transition [50]. The recovery trajectories are nonlinear, and the thresholds of intervention reversing the transition are different from the ones that maintained the previous grassland state. For example, in a hysteretic shrub-encroached grassland, low-intensity prescribed fires may have no long-term effectiveness and fail to shift the ecosystem to the previous intact grassland [19,51]. However, extreme prescribed fire under drought conditions may reverse woody encroachment by reducing the survival and density of woody resprouts [52], indicating that a "path back" exists and suggesting the possible consequences of a strong management intervention. Such grasslands exhibit hysteresis. We rated shrublands and woodlands as hysteretic to a fire treatment if they could not be reversed via the historical fire treatment maintained by the intact grassland states, but could be reversed via more intense fire treatments. Transitions were considered irreversible by fire if the fire had insignificant or short-term effects on reversing the transition. Grasslands highly encroached by resprouting trees may be irreversible by high-intensity fires due to fast regeneration [53].

### 3.1.4. Thresholds of Fire Frequency, Intensity and Severity for Shrublands

We summarized a broad range of fire thresholds without considering specific species and ecosystems from the published fire studies (Table 2). Generally, fires with a high frequency and high intensity in dry seasons can reverse woody encroachment. The fire intervals for maintaining the grasslands, shrublands, and woodland would be 1–3 years, 3–8 years, and longer than 10 years, respectively. High-intensity prescribed fires in dry seasons can surpass the mortality thresholds of non-resprouting trees. Long-term monitoring for fire impacts on woody encroachment is also needed.

**Table 2.** Fire regimes and fire thresholds for woody encroachment and their cited times in the collected studies.

| Brush Management (Cited Time) | Level 1 (Cited Time) | Level 2 (Cited Time) | Level 3 (Cited Time) |
|---|---|---|---|
| Fire type | Prescribed fires (48) | Wildfires (12) | Wildland fires (6) |
| Fire interval | 1–3 years (15) | 3–8 years (11) | >8 years (12) |
| Fire return intensity | Low intensity (2) (e.g., 627–1173 Kw/m$^2$) | / | High (7) (e.g., 23,879–68,613 Kw/Kw/m$^2$), 5291–8595 Kw/m$^2$) |
| Fire severity | / | / | Extreme severity (4) |
| Seasonality | Wet periods during the growing season (6) | Other seasons (6) | Dry seasons (7) |
| Fire recency | <one year (3) | 1–10 years (5) | >10 years (5) |
| Fire with other brush management | Fire alone (43) | Fire combined with one (21) | Fire with two other methods (2) |

Historically, low-intensity frequent fires and grazing maintained grassland states without rapid shrub and tree expansion in the Great Plains region [54]. The fire frequency thresholds are context-specific, depending on the life-history traits of grasses and encroaching species (Table 3). For example, a 5-year fire cycle protected the fescue grasslands from serviceberry (*Amelanchier alnifolia*) and cherry (*Prunus* spp.) encroachment in the Canadian prairie [55], while a fire return interval of less than 3 years was needed to maintain the tallgrass prairies in the Central Great Plains [27].

Once shrublands reach a large size, hysteresis means that the process of reversing shrublands to grasslands is not as simple as restoring the historical fire frequency. As a result, prescribed fires that only restore the historical fire frequency may fail to control shrub encroachment [51,56]. Annual burning may delay, but not reverse, the shrubs in tallgrass prairies in the Central Great Plains, as encroachers replace fine fuels, keeping fuel loads too limited to cause the high-intensity fires needed to kill encroachers [27,57]. Hysteresis has also been observed for resprouting shrubs. In the fescue grasslands in the Canadian prairies, prescribed burning that restored a 5-year fire cycle did not change the cover of serviceberry (*Amelanchier alnifolia*) and cherry (*Prunus* spp.) [55]. Also, annual burnings decreased the shrub covers but increased the sucker covers [58]. In the Northern and Southern Great Plains ecoregions, prescribed burning alone during any point in the annual growth cycle may result in little more than 30% mortality of resprouting salt cedar (*Tamarix* spp.), as this species vigorously regenerates from the roots [59].

To shift the shrublands back to grassland dominance, fire intensities that are sufficient to cause the mortality of encroaching shrubs may exclude shrub plants and result in hysteretic responses, especially amongst obligate seeding species [53]. High-intensity fire experiments have been conducted in the Great Plains region to explore fire-mortality thresholds and manage shrub and tree encroachment [60]. For example, dense canopies of juniper woodland in the Central Great Plains can intercept sunlight, suppressing grass growth, causing decreases in fuel and fire transmission and creating a feedback loop between juniper and fire suppression [27]. The reintroduction of annual burning in grasslands does not necessarily reverse juniper woodland, as the seedlings were re-established within 1–2 years and reach a similar density to the unburned areas in 5–11 years [61]. However, a fire intensity threshold of 160 KJ m$^{-1}$s$^{-1}$ caused all the juniper trees to be completely scorched and killed [53]. Prescribed burning exceeding this intensity threshold successfully shifted juniper woodlands to grasslands [62].

Fire intensity thresholds for sprouting species are less known, and reversing woody encroachment for such species is more challenging than for obligate seeders and may reflect irreversibility. Sprouting species can persist through fire events by protecting buds below ground and resprouting following fire [63], making the reversal of woody encroachment difficult. High-intensity fires can lower the shrub resprouting density during drought in the growing season, but the rates of mortality vary within and among species with

uncertain drivers of these variations [52,60]. However, it is difficult to control fire behaviour and cause fires of adequate intensity that could result in shrub or tree mortality due to undesirable weather, topography, and fuel properties. Choosing appropriate weather and topography conditions (e.g., through varying seasonality and slope) and/or manipulating fuel properties (e.g., removing grazers to increase fine fuel loads or targeting low fine fuel moisture and other environmental conditions) can drive fire intensity above the mortality thresholds of shrubs, helping to meet the restoration objectives [53,64].

**Table 3.** Thresholds of fire frequency, intensity, and severity for shrub encroachers.

| Shrub Encroachers | Fire Return | Intensity/Severity | Resilience |
|---|---|---|---|
| Clonal and resprouting shrubs (*Cornus drummondii, Rhus glabra,* and *Prunus americana*), the Central Great Plains | 1- to 3-year fire returns [27] | Annual burning can delay, but not reverse, shrub encroachment in tallgrass prairies [57]. 160 KJ m$^{-1}$s$^{-1}$: may exclude juniper encroachment [53]. | Hysteresis |
| Resprouting serviceberry (*Amelanchier alnifolia*) and cherry (*Prunus* spp.), Canadian Prairies | A historical 5-year fire cycle [55] | A 5-year fire cycle: no change in shrub cover [55]. Annual burnings: decreased shrub cover but increased sucker cover [58] An extremely severe wildfire: no change in shrub covers due to resprouting [6]. | Irreversibility |
| Resprouting salt cedar (*Tamarix* spp.), the Northern and Southern Great Plains [59] | Context-specific with historical fire frequency in specific locations | Annual: no more than 30% mortality due to sprouts [59]. A high-severity fire: large top-kill but rapid resprouting [65]. | Irreversibility |

### 3.1.5. Thresholds of Fire Frequency, Intensity and Severity for Woodlands

Greater fire suppression is required to enable woodland transitions rather than shrubland formation. Unlike shrubs, trees generally require a long time to establish and suffer high mortality rates from hot fires in their early years. For instance, shrubs can be present in tallgrass prairies after 30 years of prescribed fire at 3- to 4-year frequencies, but not trees [66]. Fire intensity and frequency thresholds vary with tree species, due to various physiological thresholds of fire resistance among encroachers, grass-woody competition, and their interactions with the local environmental drivers. For example, eastern red cedar (*Juniperus virginiana*), exhibiting the greatest tree expansion in the Central Great Plains, can resist typical grassland fires when they reach a height of 2.5 m and a diameter at breast height of 17.5 cm [27,33]. Low-intensity fires have the potential to kill immature trees, eliminating the potential local seed source of non-sprouting species [67]. Fire returns >8 years allowed eastern red cedar to encroach into grassland, and fire-free intervals of 15–20 years or complete fire suppression allowed consistent tree establishment and canopy closure. However, it is difficult to detect continuous thresholds of fire suppression for woodland transitions due to the long lifespan of woody species and the abrupt historical shift of fire regimes from frequent fire to complete fire suppression.

Manipulating fire frequency is insufficient, but surpassing the fire intensity-mortality thresholds is likely to reduce the woody-dominant states that have non-sprouting encroachers, reflecting hysteresis (Table 4). It is more challenging to use fire to reverse the encroachment of resprouting species than it is for non-sprouters. High-intensity fires have the potential to reduce tree encroachment by causing mortality and reducing resprouting but it is difficult to reverse this transition. For example, aspen, a fire-adapted species that can regenerate from vegetative sprouting and seeds, has encroached into the fescue grasslands of the Canadian prairies. Aspen has thin bark, and even low-intensity fires can cause mixed- or high-severity effects. However, as a resprouting and clonal species, aspen regenerates prolifically via vegetative sprouting after fires, and the mineral soil exposed

to high-severity fires may also favour aspen reproduction as it has long, silky seeds that disperse long distances [68]. Annual burning in different seasons from 1975 to 2010 did not reduce aspen cover in Saskatchewan, Canada [19]. Aspen canopy, aspen regeneration, and shrub expansion did not decline significantly after prescribed burning of native fescue grasslands [55]. With low-, moderate-, and high-severity fires, aspen sprout density and growth rates increased [69]. Resprouting mesquite species (*Prosopis* spp.) have increased in dominance and abundance in the Southern Great Plains. High-intensity fires caused mesquite top-kills and epicormic bud loss, reducing the resprout numbers, but did not overcome mesquite persistence in the long term [22,23].

**Table 4.** Thresholds of fire frequency, intensity, and severity on reversing woodland transitions in the Great Plains region.

| Tree Encroachers | Fire Returns | Intensity/Severity | Resilience |
|---|---|---|---|
| Non-resprouting *Juniperus virginiana* and *Juniperus scopulorum*, Northern and Central Great Plains [1] | >8 years allowed encroaching [53,64]; 15–20 years or complete fire suppression allowed close-canopy formation [1] | Fire intensity threshold of 160 KJ m$^{-1}$s$^{-1}$ [53,64]. Extremely prescribed burnings that surpassed 160 KJ m$^{-1}$s$^{-1}$ caused 100% juniper mortality and successfully shifted juniper woodlands to grasslands [62]. | Hysteresis |
| Resprouting and clonal aspen in fescue grasslands, Canadian prairies | Fire suppression | As fire intensity increased from low-, moderate-, to high, aspen sprout density and growth rates increased [69]. | Irreversibility |
| Resprouting mesquite, the Southern Great Plains | Fire suppression | High severity fires: top-kills, epicormic buds loss, and reduced resprout numbers but in the short-term [22,23]. | Irreversibility |
| Resprouting and clonal eastern cottonwoods (*Populus deltoides*) in the mixed-grass prairies in the Southern Great Plains | Fire suppression | High-severity fire: high mortality but resprouting [60]. Browsing: reduced resprouting cover [65]. | Irreversibility |

Woodland transitions exhibit hysteresis and irreversibility by fire under different stages. Fire suppression leads to tree invasion and woodland transitions, and these two hysteretic stages with non-sprouting species can only be reversed via frequent and intense fires. However, woodlands dominated by resprouting species typically cannot be reversed using fire alone. Additional management strategies, such as the costly physical and chemical removal of encroaching tree species, are necessary [27,70]. Some reversible thresholds have been wrongly hypothesized as irreversible, based on a limited range of variability in fire intensity, and sometimes these supposedly irreversible thresholds represent social–political constraints rather than ecological thresholds.

### 3.2. Brush Management Using Fire in Grassland National Parks

3.2.1. Fire Management of Woody Encroachment in Canadian and USA National Parks

In cooperation with other federal and local agencies, Parks Canada and the US National Park Service are responsible for brush management and fire management in their respective parks. National parks in Canada and the USA utilize ecological monitoring indicators to assess the integrity of grassland ecosystems [71,72]. Indicators related to woody encroachment include vegetation composition (e.g., native grass, non-native grasses, shrubs, or trees), vegetation structure (e.g., cover, height, and landscape dynamics), fire disturbance, and the occurrence and habitats of native wildlife affected by woody encroachment.

Divergences of indicator scores from expected values have been assessed as an acceptable variation, potential management concern, or imminent loss based on the resistance,

hysteresis, and irreversibility of the indicator [72]. For example, if returning to the pre-transition fire regime is generally sufficient to maintain the previous grassland state, park managers would identify the grasslands as being within an acceptable range of variation. Otherwise, hysteretic and irreversible responses would raise potential management concerns and indicate a risk of imminent loss.

Prescribed fires and wildfires have played some roles in protecting native grasslands, but in some cases have not been powerful enough to fully reverse shrublands and woodlands in parks [55,73]. Mechanical, chemical, and biological treatments have also been used in national parks to protect grassland ecosystems from woody encroachment and invasive species [19,55,73,74].

3.2.2. Case Study: Grassland Transitions to Non-Native Prairies and Shrublands

The Grasslands, Badlands, and Elk Island National Parks represent a well-preserved grassland state, a degraded grassland state, and a transition to the shrubland state, respectively (Table 5). The Grasslands National Park's iconic mixed-grass prairie is well-preserved, and only slightly threatened by agricultural grass species rather than shrubs or trees [48]. In Badlands National Park, the non-native Kentucky bluegrass (*Poa pratensis*) dominates the non-native prairie–Kentucky bluegrass plant community and only a few native forbs or shrubs are seen [75]. In Elk Island National Park, shrubs have encroached into the fragmented fescue grasslands [76].

In practice, wildland fires, including wildfires and prescribed fires, can maintain well-preserved and degraded grassland states. Even under wildfire suppression, prescribed fires may be sufficient to control the non-native species in the Grasslands National Park, and park managers have set the goal to restore the areas disturbed by invasive species and prevent new invasive plants by 2030 [48,77]. However, once non-native species become dominant or shrubland transitions occur, degraded grassland states cannot be reversed by varying the fire frequency alone. For example, in the Badlands National Park, fires only modestly reduced the non-native grass cover in non-native grasslands [78], and the years since fires have had no significant impact on the richness of native species and relative cover of exotic species [75]. Generally, high-frequency fires are needed to reverse shrub encroachment [27], but the fire cycle in Elk Island National Park is not sufficiently restored yet, resulting in the unsuccessful management of shrub and aspen encroachment [76].

The effectiveness of these management approaches varies due to ecological and social–political constraints. Firstly, fire suppression still occurs and the high-frequency fires needed to halt shrubland encroachment cannot be ensured. Fire suppression was implemented from 2002 to 2022 in Elk Island National Park due to operational and social constraints, such as undesirable weather conditions, inadequate funding, or COVID-19 [79]. Secondly, ecological interactions between fires and grazing, climate, and other disturbances are complex, making the fire thresholds unknown. For example, precipitation has more influence on vegetation cover than fire does in the Northern Great Plains grasslands, and vegetation response to climate is less predictable [75]. Thirdly, operational constraints also limit the effectiveness of combined management actions on shrub encroachment. Bison grazing and the creation of bison wallows can limit the cover of dominant invasive grass species in mixed-grass prairies [80]; however, a population of 700 bison in Badlands National Park has had little effect on the grass community due to their relatively low density [78].

**Table 5.** Fire management efficiency and recommendations on controlling shrubland transitions in selected grassland national parks.

| Alternative States | National Parks | Fire Regimes | Fire Management Effectiveness | Management Recommendations |
|---|---|---|---|---|
| Grassland states | Grasslands National Park: invasive grass [48] | Prescribed fires, a 5-year fire cycle since 2000 [48], Suppressing all wildfires [77] | Effective to maintain the grasslands [48] | Reintroduced bison in 2005 and uncleared impacts [48] |
| Shrubland transitions | Elk Island National Park: shrubs and aspen [76,79,81] | Prescribed fires since 1979 [82]; Fire suppression from 2002 to 2022 [79] | Not effective, due to fire suppression | Restore the fire cycle [76] |
| | Badlands National Park: non-native grass formed a near-monoculture | Historical fire return 8–25 years [78]; Prescribed fire since the 1980s [83] | Fire alone cannot reverse non-native grass [75] | Combinations of native seeding, fire, and/or herbicide [78] |

### 3.2.3. Case Study: Woodland Transitions

Three USA national parks and three Canadian national parks were selected as having representative grasslands that are being encroached by typical woody encroachers, such as ponderosa pine, Ashe juniper, red cedar, mesquite, and aspen (Table 6). The Lyndon B. Johnson National Historical Park is heavily invaded by exotic grasses, forbs, and the native honey mesquite in the mixed grasslands [84]. Ashe juniper and red cedar have rapidly spread throughout the tallgrass and mixed grasslands due to fire suppression in the Chickasaw National Recreation Area [85,86]. The Lake Meredith National Recreation Area also features open stands of mesquite in the prairies [87]. The Riding Mountain, Prince Albert, and Waterton Lakes National Parks in Canada were selected to represent the fescue grasslands being encroached by sprouting aspen woodlands [5,88,89].

Despite efforts to reduce woody encroachment by obligate seed species via prescribed burning and wildfires, encroachment remains a persistent challenge in these parks. Social–political constraints may be responsible for the failure of fires to reduce the woody encroachment of non-sprouting species or mature trees of sprouting species. The frequency and intensity of fires have not reached the fire thresholds of these encroachers at the three USA sites. For example, in the Lyndon B. Johnson National Historical Park, prescribed fire was only used in 2015, due to the urban location of the prairie in need of restoration [84,90]. This social constraint is challenging. Efforts have been made to increase the intensity of prescribed burns, such as by shifting the timing of the burns, to control mesquite in the Lake Meredith National Recreation Area, but this approach faces challenges due to safety considerations and the low postfire recovery of grasslands under extended drought conditions [87]. In the Southern Great Plains network, intense fires can kill large juniper trees, and subsequent fires within 10 or 15 years have the potential to kill immature junipers [86]. Intense fires might therefore be effective in reducing the encroachment of non-sprouting species in the Chickasaw National Recreation Area. However, the uncertainty associated with the use of fire and vegetation responses is still being considered by fire managers [86].

The situation is different for sprouting species in the three Canadian national parks. Even when prescribed fires and high-intensity wildfires occur, sprouting species encroachment cannot be reversed due to their rapid postfire recruitment. This suggests the presence of ecologically irreversible fire thresholds. Frequent prescribed fires in the Waterton Lakes National Park, aiming at restoring the historical fire regime that maintained the fescue grasslands, were insufficient to reverse, or even halt, shrub and aspen encroachment. The areas of aspen canopy (aspen higher than 2.5 m), aspen regeneration, and shrubs did not change significantly before and after the prescribed fires [55]. Aspen cover still expanded into burned areas in the Prince Albert National Park, indicating that prescribed fires there had limited long-term effectiveness in controlling aspen encroachment [19]. Similarly,

a high-severity wildfire can halt but not necessarily reverse woody encroachment. The Kenow fire caused the mortality of mature aspen and altered the aspen stand structure from a late-seral to an early-seral state one-year postfire; however, the extent of aspen cover remained the same, as the fire stimulated vigorous aspen sprouting [6]. In the Riding Mountain National Park, fire alone may be insufficient to suppress aspen, and a combination of brush management and prescribed fires has been recommended [73].

In places, such as in the wildland–urban interface or where irreversible ecological fire thresholds exist, a comprehensive approach that combines fire management with other brush management methods is necessary. This approach should take into account the unique challenges posed by these environments and strive to minimize the risk of catastrophic fire events while balancing ecological concerns and social needs. For example, in the Southern Plains network, fire managers are monitoring the effects of wildfire, prescribed fire, and mechanical treatments on the ecosystems [90]. In Waterton Lakes National Park, high-severity prescribed fires (e.g., applying prescribed fires in late season to utilize sufficient fuel to generate high-severity fire) and the use of Traditional Ecological Knowledge (TEK) (e.g., bison dynamics) have been recommended if management is to better conserve the grasslands [6]. However, many uncertainties and knowledge gaps exist regarding fire management and grassland conservation in these national parks. The detailed thresholds of fire frequency and severity/intensity, the interactions of fires with other disturbances, and the effectiveness of these brush management methods on woody encroachment are unknown, and long-term monitoring is needed in practice.

**Table 6.** Fire management efficiency and recommendations on controlling woodland transitions in selected grassland national parks.

| National Parks | Fire Regimes | Fire Management Effectiveness | Management Recommendations |
|---|---|---|---|
| Riding Mountain National Park: Aspen [5] | Prescribed fires (1996–2010) | Fire alone may be insufficient to suppress aspen | Combination of brush management [73] |
| Prince Albert National Park: Aspen and shrubs [88] | Prescribed fires since 1975, a 5-year fire returns 2018 Rabbit Creek Wildfire | Insufficient, limited long-term effectiveness [19] | Interactions between fire and grazing [19] |
| Waterton Lakes National Park: Aspen [91] | Prescribed fires since 1989, a 5-year fire cycle; 2017 Kenow Wildfire [89] | Prescribed fires have no impacts [55]; Wildfire was insufficient due to vigorous sprouting [6] | Improving fire severity in late season, bison [6] |
| Lyndon B. Johnson National Historical Park: Native honey mesquite [84] | Prescribed fire in 2015 [84,90] | Unknown | Restoring fire cycles, high-intensity fires |
| Chickasaw National Recreation Area: Ashe juniper, and red cedar [85,86] | Prescribed fire since 1998; several incidents of wildfires | Insufficient, due to resprouting | Mechanical removal of junipers in combination with fire treatments |
| Lake Meredith National Recreation Area: mesquite [87] | Prescribed fires since 1998, several incidents of wildfires | Insufficient, reduced invasive grass, but not woody encroachment | High-intensity, shifting the timing of the prescribed burns |

### 3.2.4. Social–Political Constraints of Resilience-Based Fire Management on Woody Encroachment in National Parks

Fire experiments conducted in the Great Plains region and North America have demonstrated the resistance, hysteresis, and irreversibility attributes of grasslands when reversing woody encroachment by fires. Grasslands in some national parks demonstrated their resistance and irreversibility, but in other parks, the hysteretic shrubland and woodland transitions were irreversible using fire (Table 6). In the early stages of woody encroachment, intact grasslands exhibit resistance to shrub encroachment, and the prescribed fires that restore historical fire frequencies may help to maintain the grasslands within the acceptable range of variability. Once the shrublands or non-sprouting woodlands are

established, the grasslands may exhibit hysteresis, creating management concerns and challenges. High-intensity fires that exceed the fire-mortality thresholds are necessary to exclude encroachment, but social–political constraints often hinder fire management efforts. In cases where resprouting species have encroached on grasslands, the grasslands may become irreversibly altered, signaling imminent loss. While fire can halt encroachment, it cannot reverse the transitions caused by resprouting species. Other postfire programs, such as encouraging elk browse on aspen regeneration or bison reintroductions, may be necessary for brush management.

Grassland resilience to woody encroachment is lower than ecologists expected in some parks due to social–political constraints (Table 7). Fire suppression, COVID-19, inadequate funding, and difficulties in implementing prescribed burning (e.g., fire danger near urban locations, and undesirable weather conditions) have all hindered efforts to reverse shrublands using fire. Furthermore, the uncertainty associated with interactions between fire and drought under climate change has further complicated the fire management efforts. For irreversible woodland transitions with resprouting species, these social–political constraints for fire management still exist, especially the unknown effects of combination management, including unknown ecological interactions and a lack of long-term monitoring. Table 7 compares the theoretical grassland resilience to woody encroachment observed in the Great Plains region to the operational effectiveness of fire management in the selected grassland national parks in Canada and the USA.

**Table 7.** Theoretical grassland resilience to woody encroachment and the operational effectiveness of fire management in selected grassland national park.

| Vegetation States | Intact Grasslands | Shrubland Transitions | Woodland Transitions: Non-Sprouting | Woodland Transitions: Resprouting Species |
|---|---|---|---|---|
| Grass resilience | Resistance | Hysteresis | Hysteresis | Irreversibly |
| Monitoring ranking in national parks | Acceptable change in variation | Fire management concerns | Fire management concerns | Imminent loss |
| Management effectiveness | Success | Limited effects | Limited effects | Limited effects |
| Grass resilience in practice in national parks | Resistance | Hysteresis and irreversibility | Irreversibly | Irreversibly |
| Social–political facilitations/constraints in national parks | (a) Restored fire frequency (b) Implementation of fire and bison | (a) Fire suppression (b) The uncertainty of fire interactions with other disturbances and climate change (c) COVID-19 (d) Inadequate funding (e) Implementation difficulties of frequent or intensive prescribed burnings, such as undesired weather | (a) Fire danger near urban locations, (b) Unexpected drought conditions increasing fire danger (c) Uncertainty associated with the use of fire and vegetation responses. | (a) Ecological irreversibility due to postfire resprouting (b) Social–political constraints that limited reversing shrublands (c) Challenges of combination management: unknown ecological interactions, lack of long-term monitoring, and unclear management effectiveness |

### 3.2.5. Resilience-Based Management Framework for Woody Encroachment

The National Park systems in Canada and the USA have established monitoring indicators for grassland ecosystem integrity and ranked each indicator as within an acceptable range of variation, a management concern, or an imminent loss. Here we focused

specifically on woody encroachment and introduce a fire management framework that is grounded in the resilience theory (Figure 3). Initially, the alternative states of grassland subject to woody encroachment would be identified according to the shrub and tree species. Next, the park's monitored grassland responses to woody encroachment via fire management reveal resilience characteristics, including resistance, hysteresis, and irreversibility under each potential state. If no management intervention or reverting to the original fire regime is sufficient to retain the previous grassland state, it suggests resistance. On the other hand, the existence of hysteresis is indicated when the vegetation transition pathway from an altered state back to grassland is different from the pathway of grassland transitioning to the altered state. When irreversible fire thresholds segregate the potential grassland states, this signals an imminent risk to the ecosystem, pointing toward management failure.

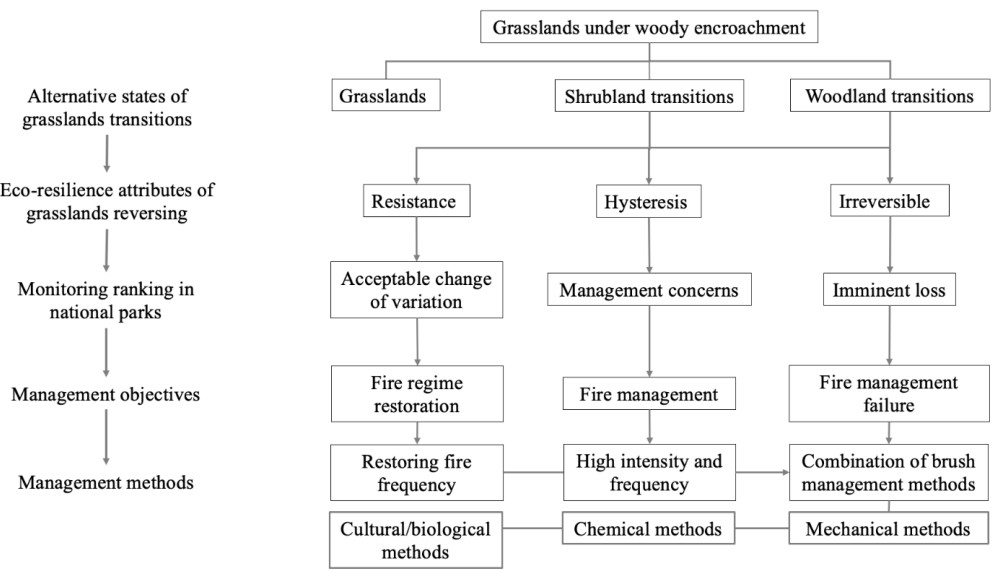

**Figure 3.** Proposed ecological resilience-based management framework for woody encroachment in Canadian and USA grassland national parks.

Identifying key indicators that adequately represent the grassland responses and resilience characteristics is challenging. Vegetation composition and structure serve as direct indicators, while wildlife populations and disturbances can be seen as indirect indicators of woody encroachment within parks. Further, the status of each indicator is classified as either an acceptable change, potential management concern, or imminent loss, based on the resilience attributes of resistance, hysteresis, and irreversibility. To manage resistant, hysteretic, and irreversible grasslands, the restoration of the fire regime is required. This may entail frequent controlled burns, intensive fires, or a combination of brush management strategies.

## 4. Conclusions

We synthesized the fire thresholds required to control woody encroachment across various shrubs and trees in the Great Plains region. We found that grassland states are inherently resistant to woody encroachment, and frequent fires alone can maintain these states before the establishment of shrubs. However, once shrublands are established, high-intensity fires that exceed the fire-mortality thresholds are required to remove existing shrubs and to exclude further encroachment, reflecting a state of hysteresis. We have also observed that fire alone will not necessarily reverse the encroachment of resprouting and clonal species into grasslands, reflecting irreversibility via fire management. Fire management combined with other brush management, as well as considering the impacts of climate-fire interactions on woody encroachment, has been widely recommended for hysteretic and irreversibly encroached grasslands. While we did not consider all encroaching



species and just focused on the characteristics of fire regimes in our selected studies in the Great Plains region, our synthesis could provide insights into the complex dynamics of alternate states of grasslands threatened by woody encroachment. These insights might then serve as a foundation for developing effective management strategies aimed at conserving a desirable state of disturbed ecosystems.

We evaluated the effectiveness of brush management using fire and proposed a resilience-based management framework for woody encroachment. In national parks, intact grassland states may be maintained via historical fire frequencies, while high-intensity fires above the tree mortality thresholds have the potential to reverse hysteresis shrubland and woodland transitions. However, social–political constraints limited the effectiveness of fire management on hysteretic shrubland and woodland transitions with non-sprouting species. Moreover, among these various states, reversing the sprouting woodland transitions proved to be the most challenging. This difficulty arose due to the existence of irreversible ecological fire thresholds and social–political constraints, both of which significantly impede the efforts to reverse the transition. Although we focused on only nine parks containing representative grassland states, our study could provide a valuable tool for identifying early signs of fire management success or failure and for the development of integrated brush management plans in grassland national parks or similar protected areas.

**Supplementary Materials:** The following supporting information can be downloaded at: https://www.mdpi.com/article/10.3390/cli11110219/s1.

**Author Contributions:** Conceptualization, H.L. and G.W.; methodology, H.L., G.W. and W.W.; formal analysis, H.L. and A.S.; data curation, H.L. and A.S.; writing—original draft preparation, H.L.; writing—review and editing, G.W., J.L.I., W.W. and A.S.; supervision, G.W., J.L.I. and A.S.; project administration, G.W.; funding acquisition, G.W. All authors have read and agreed to the published version of the manuscript.

**Funding:** This research was supported by the UBC-APFNet National Park Research Grant GR025939.

**Data Availability Statement:** All datasets presented in this study can be found within the article and in the Supplementary Materials.

**Conflicts of Interest:** The authors declare no conflict of interest.

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
