# Peer review of "Grassland Resilience to Woody Encroachment in North America and the Effectiveness of Using Fire in National Parks"

_climate, doi:10.3390/cli11110219_

Round 1

Reviewer 1 Report

Comments and Suggestions for Authors

I have marked my corrections and suggestions about the article in the text and added them as notes. If the authors take these suggestions into consideration, it will contribute to the publication of the article in a more scientific way.

Author Response

Thank you very much for taking the time to review this manuscript. 

Please find the detailed responses below and the corresponding revisions in track changes in the re-submitted files.

Comments 1:

When the necessary explanations are made in the text, there is no need to give this flow diagram, so it can be removed.

Response:

Thank you for pointing this out. We agree with this comment.

Therefore, we have updated the whole section on Materials and methods (page 3, lines 114-205) and the research questions (the last paragraph in the Introduction, pages 2-3, lines 81-113).

Comments 2:

It would be useful to provide information about fire severity, fire size, burned area scale, threshold values and limits so that readers can get a more comprehensive and satisfying information.

Comments 3: range ler verilmeden anlaşılmıyor

Response:

Agree.

To better demonstrate fire impacts on woody encroachment, we deleted the general introduction of fire regime history in the Great Plains at a large scale (page 8, lines 321-337) and focused on the specific fire regimes and fire management used in sites (figure 2). We first defined the definitions of fire regimes and then summarized how these fire regime characteristics were used in the specific studies we collected (table 2).

Comments 4: If this is not the journal's referencing convention, it would be more practical to give an abbreviation such as (NRCS, 2017) in the citation, as in "Natural Resources Conservation Service, 2017".

Response: Modification in the line 297.

Comments 5: TThe spelling of "The..." in L347 should be corrected.

here there is an extra space before the word, it should be removed.

Response:

Agree. The paragraphs were removed due to framework modification.

Comments 6:

bilimsel isimler ltince verilmelidir

Scientific names of species should be italicized. There are similar errors or misspellings in the text.

it should be italic

Response:

Agree.

Modifications in lines 186-187, 249-255, 622, 644, 645, 652. Tables 3 and 4.  

Comments 7:

it would be useful to give other fire intensity thresholds like this one,

it would be useful to share the fire intensity data here

Is there any threshold data here?

Response:

Agree.

We provided other fire intensity data or thresholds in Table 2.

Fire intensity, especially high intensity was also difficult to achieve in fire experiments. We collected 66 case studies from the literature review, and case studies of fire intensity were less than 10 (figure 2).  

The intensity data from some cited papers was not always available and just described the impacts on vegetation, such as mortality, therefore, I modified them as fire severity defined in lines 336-337. We also modified the intensity as intensity and severity in the tables (Tables 3 and 4) and subtitles (3.1.4, 3.1.4).

Comments 8: some of the words in the columns of the tables are throwable in the bottom row, it would be more organized to give them in the bottom row.

Response:

Agree.

We reorganized tables 3, 4, 5,6, 7.

Comments 9:

It should be emphasized here that aspen seeds can fly in from long distances, and that aspen can come into the area with seeds after fires that expose mineral soil.

Response:

Agree. We added it in line 727.

Comments 10:

Some of the words in the text in the table have different sizes and fonts, and should be checked and corrected.

Response:

Agree.

We reorganized tables 3, 4, 5,6, 7.

Comments 11: Unmarked set by MDPI

Response:

Agree. We updated on Page 17, Line 814.

Response to Comments on the Quality of English Language

Response:

I tried to refine most paragraphs. The detailed responses are in the attached re-submitted fire.

Additional clarifications

The framework of this reversion was adjusted to make it more organized and readable.

Also, I changed to focus more on the collected studies rather than general descriptions of the Great Plains to summarize the encroachers, fire regimes and thresholds, and fire impacts on woody encroachment.   

Reviewer 2 Report

Comments and Suggestions for Authors

The subject of this paper is of great interest globally with so many savannas and woodlands threatened by woody encroachment due to fire suppression. I agree entirely with the invocation of hysteresis refers to the use of only fire to restore encroached open ecosystems, but I do think that the idea of ‘irreversibility’, as used in this manuscript, deserves further clarification. I believe that word is usually used to refer to reversals with the sole use of fire, given that with sufficient heavy equipment and herbicides, often accompanied by fire, even dense forest can be converted into savanna.  This switching of ecosystems, from forest to savanna at least, is given a lot of attention internationally, but few researchers focus on there verse process.

I also wonder why a manuscript of this sort was submitted to a journal called “Climate”?  There are obvious climate implications of woody encroachment and its reversals, but those effects are not at all discussed in this manuscript. I wonder, for example, about carbon stocks as well as albedo effects of conversion of open ecosystems to woodlands and forests.

Throughout the manuscript, due to my ignorance of the natural history of the species mentioned, I kept wondering whether each species that sprouted was also capable of vegetative spread (i.e., clonal spread). I know that aspen spreads vegetatively, often to great extents, but I don’t know about the others and I think the distinction is very important. Once an area is colonized by clonal woody species, I start wondering about the hysteresis issue.

I often struggled to decipher exactly what the authors meant such as on lines 50-51: “may face greater variability in feedback loops.” I’m sure the authors know exactly what they mean, but I am not sure of the sorts of variability being considered as well as about the nature of the ‘feedback loops’---which I believe can be positive or negative. This phrase comes at a critical point in the manuscript when terms and processes are being defined, so it needs to be absolutely clear. Also see line 57—same problem, ‘full range of variability in feedback loops’.

Throughout the manuscript there are extra words that keep the prose from being easy to read. The word ‘significant’, for example, is used excessively (e.g., line 67 ‘significant’ grasslands). Also, doesn’t ‘fire management implementation’ mean the same as ‘fire management’?

When the sites were first mentioned I wanted to see a map, but it comes only much later and then isn’t at all what I needed….I am not familiar with the geography of the ‘Great Plains’ but this map was not very useful.

Define terms the first time they are used. What, to you, is a ‘woodland’ or a ‘shrubland’? used very differently around the world. Note that top kill is different from mortality and that stump sprouting is different from clonal expansion.

No mention of climate change, which is curious. Also, shouldn’t ‘grazing’ be differentiated at least between bison and cattle?

If Climate is an international journal, then perhaps stick with Latin binomials because the common names are less than informative. On that same issue, if this is only about the Great Plains of the USA and Canada, then what about submitting it to a regional journal such as the American Midland Naturalist?  If this is for an international audience, then the authors should learn what is known about this topic in other places including the SE Coastal Plain of the USA, Brazilian Cerrados, African savannas, and the vast grassylands of Australia. A good start might be William Bond’s book, Open Ecosystems.

Somewhere in the middle of North America, aren’t their massive problems with exotic fire favoring grasses? If not in the study area, then that should be clarified.

All over the world, fire ecologists are exploring the effects of seasons of fire….there’s even millions of dollars per year flowing into North Australia to promote early dry season burns. Why is season apparently not an issue in the Great Plains?

Comments by Line Number

120 can you really classify entire parks? Isn’t there relevant within park variation?

139 ‘grass relience’?  not sure what that means

141-146 can be condensed

152-155 odd

163 ‘once to eight times each year’---this can’t be correct. As I read the ms I become increasingly convinced that it was not critically read and edited one last time before submission, which is frustrating. It’s far too long given its content, for one thing. 

165 large fire size contrasts strongly with the Australian experience….large and relevant literature.

167 ‘wildfire eradication’??? that can’t be right

172 ‘shadow era’ doesn’t makes sense

177 give the years

187. 1-2% ???

204 what about plowing? Weren’t some of these grasslands plowed?  No mention whatsoever.

211-212 prose confusing

216-221 not clear

230 only? What about with heavy equipment and herbicides?

266 wildland fire = wildfire, right?

269 Pickett and Seagle is an odd citation for this idea, and it isn’t cited.

285-286 prose problems

294 what about the Bureau of Indian Affairs?  Not a player?

311-312 prose issues

317-322 important but confused. Isn’t it resilience and not resistance? What is ‘exhibiting resistance”? And, again, ‘irreversibility’ means irreversible only with the use of fire, right?

324-327 all very confusing…trigger a range of variability and stabilizing feedbacks?

340 ‘typical fire thresholds”???

355 ‘influenced’ is non directional and unclear

379 clarify which ones sprout and which ones both sprout and spread vegetatively

420 is aspen a clonal shrub?

425 Table…clarify which sprout and which spread vegetatively…some confusing prose.

Figure 3 and 4 are too alike to warrant both

464 important but confusing

477 what about site productivity?

497 ??

506 I don’t believe this is what you mean…perhaps only after encroachment.

Table 3 not veryinformative or clear

533 exotic species of what? Disturbed by hogs?

550 ‘to burn shrub encroachment into grasslands’---this phrase is a great example of lack of careful proofing and editing

554 is this what you mean?

561 zero fires during that period in the entire park? Amazing.

713 ‘non-sprouting woodlands’---that may mean something to you, but not to this reader.

Sorry, I ran out of time and out of steam. The manuscript seems far too long for its content. I really like the focus on hysteresis, but there’s too much hard-to-follow prose in the way of delivering the message. The authors obviously have a globally important story to tell, but need to work harder on the telling.

Comments on the Quality of English Language

Sorry but due to lack of clear definitions, the text was hard to follow. Also, there were other impediments to readability, some of which I pointed out in my review. 

Round 2

Reviewer 2 Report

Comments and Suggestions for Authors

The authors did an admirable job of revising this manuscript.